# miRNAs in Pulmonary Hypertension: Mechanistic Insights and Therapeutic Potential

**DOI:** 10.3390/biomedicines13081910

**Published:** 2025-08-05

**Authors:** Jindong Fang, Hongyang Chen, Zhuangzhuang Jia, Jinjin Dai, Fengli Ma

**Affiliations:** 1School of Basic Medical Sciences, Yunnan University of Chinese Medicine, Kunming 650500, China; m13649661060@163.com (J.F.); chenhyth@163.com (H.C.); daiijinn@163.com (J.D.); malily1688@126.com (F.M.); 2Yunnan Key Laboratory of Integrated Traditional Chinese and Western Medicine for Chronic Disease in Prevention and Treatment, Kunming 650500, China; 3Key Laboratory of Microcosmic Syndrome Differentiation, Education Department of Yunnan, Kunming 650500, China

**Keywords:** Pulmonary hypertension, microRNA, vascular remodeling, metabolic reprogramming, right ventricular dysfunction

## Abstract

Pulmonary hypertension (PH) is a serious pulmonary vascular disease. Vascular remodeling, metabolic reprogramming, inflammation, and fibrosis are all major pathogenic mechanisms in PH. MicroRNAs (miRNAs) are small RNAs, about 20–24 nucleotides long, that play important regulatory roles in biological processes, and in recent years, miRNAs have been found to potentially play a regulatory role in the pathogenesis of PH, and also serve as biomarkers and therapeutic agents for PH. However, there is still a long way to go from these experimental findings to their implementation in clinical practice. This study reviews the potential role of miRNAs in the pathogenesis of PH and suggests future applications of miRNAs in PH.

## 1. Introduction

Pulmonary hypertension (PH) is a clinical and pathophysiological condition characterized by primary or utilitarian changes in the pulmonary vasculature caused by various heterogeneous illnesses (etiologies) and different pathogenic components, leading to expanded pulmonary vascular opposition and pulmonary blood vessel pressure, followed by right cardiovascular breakdown and even fatality. At present, hemodynamic levels for the determination of PH comprise mean pulmonary arterial pressure (mPAP) > 20 mmHg and pulmonary vascular obstruction (PVR) > 2.0 Wood units (WU) [1]. As per the ESC (European Society of Cardiology) guidelines or emergency room (ER) protocols, clinical group 1 is pulmonary arterial hypertension (PAH), clinical group 2 is PH related to left coronary illness, clinical group 3 is PH related to pulmonary infection as well as hypoxia, clinical group 4 is PH related to pulmonary vein impedance, and clinical group 5 is PH related to an obscure instrument and/or multifactorial PH [2]. The pathogenesis of PH is diverse and multifactorial; for example, it can present as epigenetic, mitochondrial problems, aggravation, or fibrosis (Figure 1). Recent studies have highlighted the important role of microRNAs (miRNAs) in regulating PH-related gene expression.

MicroRNAs (miRNAs) are short RNA molecules, 19 to 25 nucleotides in size, that regulate post-transcriptional silencing of target genes [3]. A single miRNA can target many mRNAs, which post-transcriptionally control gene expression and impact numerous natural cycles, including cell proliferation, apoptosis, and irritation. These cycles are closely related to the development of PH. For instance, dysregulation of miRNAs can prompt vascular remodeling and an expanded proliferation of pulmonary vascular smooth muscle cells, both of which contribute to the pathogenesis of PH [4]. Understanding the role of miRNAs in PH is essential to elucidate their complex pathogenesis and to formulate therapeutic approaches. Recent studies have recognized explicit miRNAs that are communicated differently in PH patients in contrast with control subjects, suggesting their potential as biomarkers and therapeutic targets [5]. In addition, therapeutic modulation of miRNAs has emerged as a promising intervention strategy with the potential to reverse or halt the progression of PH by targeting the underlying molecular mechanisms. One method is to intervene in the processing of RNAs by targeting, for example, pri-mRNAs and pre-miRNAs [6]. This review explores the pathogenesis of PH and describes the associated miRNAs. In addition, future directions for miRNA therapy are discussed.

## 2. Biosynthesis of miRNA

### 2.1. Classical Biosynthetic Pathways

MiRNA biosynthesis is a complex multistep process in which miRNA genes are translated into precursor miRNAs (pri-miRNAs) using RNA polymerase II. The pri-miRNA is then perceived and severed in the core by a microchip complex made of RNase III Drosha and its RNA-restricting protein, DiGeorge Disorder Basic Locale 8 (DGCR8) [7], which creates a handled, clasp-molded, ~70 nt long miRNA forerunner, otherwise called a pre-miRNA, with a trademark stem–circle structure. The pre-miRNA is then moved from the core to the cytoplasm using exportin 5 (EXP-5). There, the pre-miRNA is additionally handled by a second RNase III, Dicer, and its cofactors, TAR RNA-restricting protein (TRBP) and PKR protein activator (Settlement), to produce a double-stranded RNA consisting of the intact mature miRNA and its free strand (called the passenger strand), Back protein (AGO1-AGO4 in warm blooded creatures). After stacking, one miRNA strand (the guide strand) is bundled into the RNA-actuated quieting complex (RISC). Its 5′- nucleotide communicates with the MID underlying space of the Back protein to shape the last Prior miRNA complex, called the RNA-actuated quieting complex (RISC), while the other strand (the traveler strand) is quickly corrupted. The RISC/miRNA complex then attaches to the target mRNA and either hinders its interpretation or advances its debasement (Figure 2) [8]. Despite this, major miRNAs are created through the classical biosynthetic pathway, arising from non-classical biosynthetic pathways that don’t depend on the Drosha/Dgcr8 or Dicer pathways. Additionally, they produce miRNAs that can be utilitarian; however, these miRNAs are not quite the same as those produced by the classical pathway and should also be examined [9].

### 2.2. Non-Classical Biosynthetic Pathway

Cells can produce miRNAs through several pathways; they are not delivered exclusively through the classical pathway, but also through the non-classical pathway. In the classical pathway, they bypass at least one stage (usually the Dicer, Drosha, and Dgcr8 pathways) and are known as non-classical miRNAs. However, they mainly act like miRNAs.

#### 2.2.1. No Reliance on the Dicer Pathway

miR-451 accounts for 11% of all miRNAs reads in a normal fetal liver, but is markedly reduced in mutants [10]. pre-miR-451 has an unusually short 17-nucleotide stem area; it is not long enough to be successfully perceived and handled by the Dicer pathway. Measuring 23 nucleotides in length, the six terminal nucleotides of mature miR-451 span the circle area and extend into the reciprocal strand of the hair clip forerunner, a phenomenon that is not normal for miRNA restriction in the classical pathway. Hence, the likelihood that miR-451 might exhibit an uncommon method of biogenesis has been investigated, with the finding that miR-451 avoids a connection in the classical pathway and enters the RISC [11]. In spite of the fact that cleavage of the pri-miRNA continues regularly, the Dicer handling stage is skipped, and the pre-miRNA is stacked directly into the Argonaute protein [12]. The subsequent complex will, likewise, avoid the connection between the PAZ space and the 3′ end of the conventional Dicer item in the Argonaute protein [13].

#### 2.2.2. No Reliance on the Drosha/Dgcr8 Pathway in the Nucleus

Mirtron is an miRNA handled by a non-classical miRNA pathway that was first found in Drosophila melanogaster [11] and Cryptobranchus showyi [14]. The pathway does not need the Drosha/Dgcr8 microchip complex to deliver pre-miRNA; Mirtron itself contains pre-miRNA that is sheared by introns and afterwards moved to the cytoplasm through Exportin 5 to be cut by Dicer. Therefore, Mirtron is handled by shearing rather than the intricate microchip, and is later converged into the classical pathway [15].

Small nucleolar RNAs (snoRNAs) are an abundant class of non-coding RNAs that are principally utilized as guides for synthetic changes in ribosomal RNAs (rRNAs) [16]. Due to their specific structures and functions, snoRNAs can be grouped into H/ACA- and Box C/D-class snoRNAs. In particular, one specific snoRNA, ACA45, also does not need the Drosha/Dgcr8 complex for handling, yet requires Dicer. Drosha creates regular 2 nt 3′ cleavage features that are not the same as those of ACA45 stem–circle structure handling intermediates created by immunoprecipitation of Banner/HA (FH)-labeled DGCR8 and 32P-labeled essential miR-27a records or ACA45. No evidence has been observed that ACA45 is a substrate. The requirement for Drosha was further investigated by using a luciferase reporter construct. Indeed, no elevation in luciferase activity was observed after Drosha depletion, whereas the miR-19b response reported a significant increase in luciferase activity. The results suggest that ACA45 handling is independent from the Drosha/DGCR8 complex. The miR-27a antecedent was then productively handled by both FH-Ago2 and FH-Dicer immunoprecipitates through Dicer action, which could be co-immunoprecipitated with antibodies against Back proteins, showing that Dicer is required for ACA45 [17]. These observations have shown the presence of snoRNAs; however, investigating their usefulness with miRNAs remains necessary.

The use of Dicer-subordinate and DGCR8-non-subordinate short fasteners allowed miRNAs to be recognized in the examination of non-mirtronic genomic loci, the following four of which expressed significant miRNA genes: mir-320, mir-484, mir-668, and mir-344 [18]. They are primarily unique in relation to the classical pre-miRNA fastener flanking successions and the classical miRNA flanking positions are kept consistent at a specific level to guarantee that the chip accurately perceives the necessary matching conditions. Conversely, as short-fastener-determined miRNAs do not have arrangements that match microchips, they are not subject to the DGCR8 pathway [19].

Another non-classical pathway has been recognized in murine γ-herpesvirus 68 (MHV68). MHV68 contains various miRNAs that contain an arm of a pre-miRNA hair clip connected to a tRNA particle. This occurs due to the way pre-miRNAs are handled in the core by the tRNA-handling chemical tRNase Z as opposed to by the Drosha/Dgcr8 microchip complex, after which the created pre-miRNAs are processed into mature miRNAs. Dicer consequently processes the created pre-miRNAs into mature miRNAs [20]. In other words, miRNAs do not rely on the Drosha/Dgcr8 pathway and instead rely on tRNase Z and Dicer pathways, which are new pathways.

## 3. miRNAs Involved in the Pathogenesis of PH

miRNAs assume a significant role in the regulation of basic organic cycles such as cell proliferation, migration, and apoptosis [21]. Various miRNAs are involved in the pathogenesis of PH, and microRNA-146a (miR-146a) has been shown to advance vascular smooth muscle cell proliferation and vascular neointimal hyperplasia, with expanded expression of the miR-146a gene in the lungs of PAH patients and monocrotaline (MCT)-induced rodents [22]. Furthermore, in another review, miRNAs were removed from fringe plasma of PH patients (n: 44) and control subjects (n: 30) utilizing an miRNA confinement pack, and decreased expression of miR-138, miR-143, miR-145, miR-190, miR-204, miR-206, miR-208, has-miR-21-3p, and has-miR-143-3p was observed [23]. miRNAs are engaged in the pathogenesis of PH, playing a vital role (Table 1).

### 3.1. miRNAs Involved in Vascular Remodeling

Due the variation in the increase in certain factors, including in hypoxia, irritation, and oxidative pressure, the pulmonary arteriovenous framework changes, prompting pulmonary vascular remodeling and eventually advancing PH [48]. Additionally, during pulmonary vascular remodeling in patients with PH, vascular endothelial injury, vascular media hypertrophy, fringe vascular muscle fibrosis, and an expanded extracellular matrix (ECM) are often observed. Thus, the pulmonary vascular lumen becomes contracted, pulmonary conduits become blocked, and vasoproliferative plexiform injuries occur, which cause PH to develop [49]. Pulmonary vascular remodeling is a vital element of PH, which includes the proliferation of and phenotypic changes in intimal pulmonary arterial endothelial cells (PAECs) and pulmonary artery smooth muscle cells (PASMCs), as well as complicated connections including, among others, the external layer of pulmonary arterial fibroblasts (PAFs). In ongoing studies, miRNAs have been shown to be communicated in these cell types; they play a significant role in determining the aggregate of every cell type and in vascular remodeling components [50].

#### 3.1.1. miRNAs Regulating Pulmonary Vascular Smooth Muscle Cells

(1) miR-30d: miR-30d, an individual from the miR-30 family, plays a significant role in directing the pathogenesis of cardiovascular diseases such as cardiovascular breakdown, heart remodeling, and cardiomyocyte cell death [51]. In tests utilizing hereditary, pharmacological, and lentiviral intervention strategies to achieve overexpression (OE) or loss-of-capability (LOF) trial conditions in mice and rodents and in cases of ischemic atrial fibrillation, elevated miR-30d expression prevents pathological ventricular remodeling by preventing myocardial injury, cardiac fibrosis, and apoptosis, and conversely exacerbates ventricular remodeling. This affirms the advantages of miR-30d in cardiovascular remodeling. In addition, miR-30d has tentatively been shown to be available in human plasma, and elevated levels of miR-30d might be valuable for heart remodeling [52]. Indeed, the most recent study shows that miR-30d safeguards against pulmonary vascular remodeling [24]. MiR-30d successfully restrains PASMC proliferation and migration by focusing on metadherin (MTDH) and phosphodiesterase 5A (PDE5A). The overproliferation of PASMCs is one of the principal elements of pulmonary vascular remodeling and accordingly reduces vascular remodeling. Likewise, sildenafil is a promising medication for the treatment of PAH [53]. The target PDE5A, a key target of sildenafil, is also a target of miR-30d, suggesting that miR-30d is helpful in the treatment of PAH with sildenafil (sildenafil). While enhancing miR-30d may represent a therapeutic approach for PH, the specific ways that miR-30d acts on PH and the underlying molecular mechanisms are still unknown [4].

(2) miR-212-5p: In recent studies in hypoxic rats, miR-212-5p was found to be upregulated in pulmonary vascular cells. Similarly, in hypoxia-induced PH rats [54], this microRNA was also upregulated in the lungs. Based on these findings, a series of experimental studies were carried out to explore the role of miR-212-5p in this context. Firstly, miR-212-5p was upregulated in PASMCs and in the lungs of PH patients and rodents, as assessed by qPCR examination. Following this observation, in vitro cell tests utilizing the miR-212-5p antagonist (Anti-miR-212-5p) to decipher the role of miR-212-5p in PASMCs and PH revealed that miR-212-5p is an anti-proliferative miRNA in PASMCs. Endogenous miR-212-5p was also shown to have a defensive effect on SMCs with PH by SMC-specific knockout of miR-212-5p mouse strains [25]. miR-212-5p plays a role in the pathogenesis of PH; however, through what pathways and in what ways remain unclear. At present, a couple of upstream controllers of miR-212-5p expression have been identified [41,55,56].

(3) miR-340-5p: It was found that miR-340-5p might be prompted by cardiotrophin-1 (CT-1), a member of the interleukin 6 (IL-6) family that assumes a key role in cardiovascular breakdown and expanded cardiomyopathy [57]. In addition, miR-340-5p was found to intercede in interleukin 1 beta (IL-1β). IL-1β and IL-6 are pleiotropic cytokines that are associated with inflammation [58]. In recent studies, it was found that miR-340-5p may collaborate with IL-1β and IL-6 to partake in primate PAH. Further investigations uncovered that miR-340-5p repressed PASMC proliferation and migration by restraining IL-6 and IL-1β, consequently preventing apoplexy and PAH improvement in a rodent model, and that upregulation of miR-340-5p hinders the NF-κB pathway and NF-κB pathway by restraining IL-6 or IL-1β-prompted irritation. It was also found that upregulation of miR-340-5p restrains the NF-κB pathway and thus restrains IL-6- or IL-1β-prompted aggravation.

(4) miR-153: miR-153 is chiefly engaged in various pathophysiological cycles of human illnesses such as hindrance of cellular breakdown in the lungs [59], restraint of pulmonary fibrosis [60], and hindrance of angiogenesis under hypoxic conditions [61], in addition to having an impact on PAH. Recent studies have shown that pulmonary vascular remodeling under hypoxic conditions is mainly caused by the proliferation and migration of human pulmonary artery smooth muscle cells (HPASMCs) [62]. In the most recent studies, miR-153 was chosen as the target to explore the impact of miR-153 on the proliferation and migration of HPASMCs under hypoxic conditions. Cell proliferation was estimated by CCK-8 examination, and matrix metallopeptidase 2 (MMP-2) and proliferating cell nuclear antigen (PCNA) expression was estimated by Western blot analysis, wound healing (cell migration) assays, and Transwell cell migration assays. The results showed that the overexpression of miR-153 restrained the proliferation and migration capacity of HPASMCs under hypoxic conditions [27]. Subsequently, the systems were investigated, as Rho-associated coiled-coil-containing protein kinase 1 (ROCK1) and nuclear factor of activated T cells 3 (NFATc3) are both critical targets in PAH that are implicated in cell proliferation and migration in PAH [63]. Likewise, miR-153 is associated with the development of cell proliferation and migration in PAH; therefore, the connection between them was investigated. Finally, it has been shown that miR-153 fundamentally diminishes the protein expression of ROCK1 and NFATc3, highlighting the role of miR-153 in pulmonary vascular remodeling due to HPASMC proliferation and migration. However, this only shows that miR-153 targets ROCK1 and NFATc3 to some degree, and investigating the specific mechanism of miR-153 in PH remains necessary [27].

The regulation of smooth muscles is a key issue in vascular remodeling; for example, miR-221-3p can target the phosphatase and tensin homolog (PTEN) to promote the proliferation and migration of pulmonary artery PASMCs [64]; miR-143/145 and the Krüppel-like factor 4 (KLF4) targeting factor are involved in regulating phenotypic changes in PASMCs [65]; and miR-155-5p can target the suppressor of cytokine signaling 5 (SOCS5), thereby regulating the aberrant proliferation, migration, and contraction of HPASMCs [66]. Activation of the BMP/Smad signaling pathway through the downregulation of miR-1298 inhibits cell proliferation and migration and promotes apoptosis in hypoxia-treated PASMCs [67]. CircItgb5 promoted the transition of PASMCs to a synthetic phenotype by interacting with miR-96-5p and ubiquitin-like modifier-activating enzyme 1 (Uba1), further regulating pulmonary vascular remodeling [68]. These processes are involved in smooth muscle remodeling in PH vascular remodeling.

#### 3.1.2. miRNAs Regulating Pulmonary Vascular Endothelial Cells

(1) miR-27: Previous studies have shown that miR-27b represses endothelial cell proliferation and migration in Kawasaki disease [69] and also that it targets peroxisome proliferator-activated receptor γ (PPARγ) [28]. Accordingly, recent studies investigated whether miR-27b is associated with the pathogenesis of PH through PPARγ. They first showed, through Western blotting results, that miR-27b is upregulated and PPARγ downregulated in hypoxic HPAECs, suggesting that miR-27b is systemically related to PAH progression. This was not confirmed by bioinformatics examination, and a double-luciferase assay demonstrated that miR-27b inhibited PPARγ expression by targeting the PPARγ gene. In the Transwell migration chamber measurements, the migration of HPAECs was higher in the hypoxia + miR-27b inhibitor combination versus the hypoxia + pioglitazone (PPARγ agonist) combination. It was shown that miR-27b impacted hypoxia-incited HPAEC damage by targeting PPARγ [29]. In another study, miR-27a was used to treat endothelial damage via PPARγ; miR-27a was chosen because past examinations have shown that it diminishes lung PPARγ levels in a hypoxia-prompted PH model [30]. miR-27a increased in Berkeley SS mice and hemoglobin-treated HPAECs, and this proliferation in miR-27a hindered the upregulation of PPARγ, showing that miR-27a reduces PPARγ expression through post-transcriptional components both in vivo and in vitro [70], which makes up for the absence of in vivo miR-27b tests [29]. Likewise, the function of PPARγ in the pathogenesis of PH in various models and species is an issue that ought to be additionally examined, and may provide new approaches for the treatment of PH.

(2) miR-30a-5p: In colorectal and gastric disease tissues and cell lines, miR-30a-5p was updirected or downmanaged to restrain proliferation and prompt apoptosis in colorectal and gastric malignant growth cells [71]. miR-30a-5p is involved in cell proliferation and apoptosis [72]; however, its role in PH is still unknown. Consequently, an analysis was performed to study the role of miR-30a-5p in the proliferation and apoptosis of HPAECs under hypoxia. Firstly, the expression of miR-30a-5p in the plasma of PAH patients was determined via RNA extraction and reverse-transcription quantitative polymerase chain reaction (RT-qPCR). miR-30a-5p expression levels in HPAECs under hypoxic conditions were subsequently observed by RT-qPCR at 24, 48, and 72 h, and was found to be reduced. miR-30a-5p overexpression fundamentally diminished the pace of HPAEC apoptosis, and these results showed that miR-30a-5p overexpression was useful for the proliferation and restraint of apoptosis of HPAECs under hypoxic conditions [31]. This trial was not performed in vivo to confirm the immediate contribution of miR-30a-5p in PAH, and this should be investigated in future studies.

(3) miR-410: Past studies have shown that miR-410 is downregulated in pituitary gonadotroph tumors, and its overexpression prompts decreased cell development [73]. In pancreatic disease, miR-410 overexpression restrains pancreatic cancer cell growth in vitro and in vivo, as well as cell invasion and migration, and also resulted in G1/S cell-cycle arrest [74]. In cholangiocarcinoma, miR-410 treatment restrained CCA cancer development in xenografts [75]. Ongoing studies have confirmed the possible role of miR-410 in the pathobiology of PH by focusing on nicotinamide phosphoribosyltransferase (NAMPT), a controller of pulmonary vascular remodeling, after it was shown that NAMPT intercedes pulmonary vascular remodeling and that restraint of NAMPT migration is an expected useful target for pulmonary hypertension [76]. The downregulation of miR-410 in hypoxia-prompted pulmonary hypertension (HPH) was first exhibited in a hypoxia-interceded HPH mouse model, and afterwards, the transfection of hPAECs with miR-410 imitators or inhibitors was studied to show that miR-410 directs the outflow of NAMPT in hPAECs. It was also found that miR-410 controls the proliferation and migration of hPAECs and advances apoptosis. It was also tracked down shown that miR-410 controlled the proliferation and migration of hPAECs and advanced apoptosis. In the meantime, in vivo studies showed that miR-410 overexpression weakened the migration of HPH. These results suggest that miR-410 controls the proliferation and migration of hPAECs and advances apoptosis by focusing on NAMPT, likely playing a role in the pathogenesis of PH [32].

Furthermore, miRNA-31-5p is involved in spermine-induced autophagy by targeting N-acetyltransferase 8-like (NAT8L) to regulate PAECs [77]. Platelet-derived growth factor (PDGF) affects miR-409-5p expression and regulates pulmonary artery endothelial cell dysfunction [78]. In addition, overexpression of miR-150-5p attenuates ox-LDL-induced endothelial cell injury in human venous endothelial cells (HUVECs) [79], and is also involved in the regulation of endothelial cells in PH, indicating the importance of endothelial cell regulation in PH vascular remodeling.

#### 3.1.3. miRNAs That Regulate Fibroblasts

miR-124: miR-124 is a miRNA that contributes to the proliferation, obtrusive capacity, migration, and apoptosis of malignant growth cells and plays a significant role in disease in different cells [80]. miR-124 was shown to regulatorily affect cell proliferation and aggravation in synovial fibroblasts [33]. Building on previous studies investigating miR-124 in vascular remodeling in the pathogenesis of PH, miR-124 was viewed as diminished in profoundly proliferative and phenotypic fibroblasts segregated from pulmonary hypertensive calves (PH-Lies) and from human patients with idiopathic pulmonary blood vessel hypertension (IPAH-Lies). Overexpression of miR-124 restrained the proliferation and migration of fibroblasts, and restraint of miR-124 led to the unwanted proliferation and migration of fibroblasts. miRNAs regulate gene expression by binding to target sites of mRNAs and causing their degradation or translational repression. Through this, it was shown that polypyrimidine tract-binding protein 1 (PTBP1) is a direct downstream target of miR-124 in external layer fibroblasts and controls the proliferation of fibroblasts through PTBP1; however, the administrative system of miR-124 has not yet been fully examined. This study revealed that histone deacetylase (HDAC) inhibitors prompted a critical reduction in miR-124 and an ensuing reduction in PTBP1 expression. This indicates that HDAC inhibitors might play a role in the treatment of PH [81].

### 3.2. miRNAs Involved in Metabolic Reprogramming

To adjust to hypoxia and a lack of supplements, cells must reinvent metabolic pathways to meet the cell’s energy, biosynthetic, and redox needs in a cycle known as metabolic reprogramming, which is viewed as a sign of growth in cells [82]. The principal modalities are glucose digestion [83], lipid digestion [84], amino corrosive digestion [85], glutamine hydrolysis [65], and the pentose phosphate pathway [86]. Metabolic reinvention is also observed in smooth muscle cells (SMCs), endothelial cells (ECs), and fibroblasts (Lies) in PH, and this has been interpreted as a sign of PH [87]. In recent studies, miRNAs have, again, been found to be strongly associated with metabolic reprogramming in PH.

(1) miR-124: The proliferation, migration, and inflammation of disease cells, including stromal cells, in response to changes in cell digestion and the consequent adjustment of the connection between digestion, development, and irritation might be possible remedies for malignant growth. Additionally, there are metabolic changes in PH, the most common of which is the Warburg effect [88]. Fibroblasts in PH have a one-of-a-kind metabolic ensemble that is a key driver of pulmonary vascular fibrosis and vascular remodeling and represents a potential new direction for PH anti-fibrotic therapy [89]. In this regard, the former study aimed to uncover the mechanism behind the unique metabolic reprogramming observed in PH-Fibs, including those in the hypertensive vessel wall. Its findings highlight the potential of PH-Fibs as a novel method for understanding metabolic dysregulation under hypertension, and previous studies have also demonstrated that miR-124 regulates the proliferation of PH-Fibs through its direct target, PTBP1 [81]. Furthermore, pyruvate kinase M1/2 (PKM) isoform expression is regulated together by the following three heterogeneous nuclear ribonucleoproteins (hnRNPs): PTBP1, also known as hnRNPI, hnRNPA1, and hnRNPA2. It was shown that in PH-Lies, more significant levels of PKM2 compared with PKM1 are accompanied by decreased mitochondrial pyruvate carrier (MPC) and sirtuin 3 (SIRT3) expression, and MiR-124 then directs PKM2/PKM1 expression through PTBP1, which controls PKM isoform expression through particular grafting and reestablishing the inversion of the glycolytic switch in PH-Lies [34]. These findings suggest that metabolic reprogramming is controlled through the miR-124-PTBP1-PKM2 hub in PH-Lies cells (Figure 3).

(2) miR-22-3p: Insulin resistance (IR), chemical disturbances, and lipid metabolic reprogramming have been demonstrated to underscore lipid digestion dysfunction–a hallmark of PH [84]. In PH, lipid metabolism drives vascular cell proliferation and vascular remodeling. The regulation of lipid metabolism involves fatty acid transport, synthesis, and oxidation processes. In cancer and cardiovascular disease, miR-22 regulates lipid metabolism and vascular remodeling [90], and although its role in PAH is unclear, it remains a potential biomarker for PAH. Osthole was used to reprogram lipid metabolism to suppress cell proliferation by modulating miRNA-22–3p-mediated metabolic enzymes and metabolite C10:2, thus ameliorating pulmonary vascular remodeling. The central issue lies in the restraint of metabolic reprogramming compounds and metabolite C10:2. This suggests that miRNA-22-3p functions as an upstream gene to manage metabolic reinvention and advances the proliferation of PASMCs, which may be another possible therapeutic target for pulmonary vascular remodeling [35].

(3) miR-329-3p: Circular RNAs (CircRNAs) are covalently shut RNA atoms shaped by the inverted grafting of numerous exons or introns [91]. In PH, some circRNAs are associated with the proliferation and migration of PASMCs, indicating that they are significant controllers of PAH. The downregulation of SPARC circular-related modular calcium-binding 1 (circSMOC1) is related to a profoundly proliferative aggregate and metabolic reprogramming of PAH PASMCs. The original atomic components of metabolic reprogramming in PAH PASMCs have also been investigated [36]. It was first found that circSMOC1 is involved in metabolic reprogramming of PASMCs in PAH, and subsequently, it was shown that the downregulation of circSMOC1 in pulmonary artery smooth muscle cells promotes cell proliferation and the activation of aerobic glycolysis via the circSMOC1/PTBP1 (nucleus) and circSMOC1/miR-329-3p/PDHB (cytoplasm) pathways. Overexpression of circSMOC1 can reverse aerobic glycolysis, demonstrating its critical role in modulating metabolic abnormalities and the pathogenesis of inductive pulmonary hypertension in PASMCs. These findings provide new evidence for other relevant fields. Through the overexpression of circSMOC1, metabolic abnormalities and related pathological changes in PASMCs linked to pulmonary hypertension are effectively alleviated. This study aims to investigate the relationship between metabolic disorders in PASMCs and the pathogenesis of pulmonary hypertension. These studies suggest that miR-329-3p is involved in the metabolic reprogramming of PASMCs in PAH rats, but there are no data on human circSMOC1 expression and metabolic functions to understand its specific mechanism in humans, and further studies are needed [36].

(4) miR-125a-5p: Reduced expression of miR-125a-5p is a common feature in tumors. In recent studies, miR-125a-5p has been shown to be involved in the regulation of cancers, including gastric, liver, and laryngeal cancers, affecting the proliferation and apoptosis of tumor cells [92,93,94]. Significantly, under all oxygenated conditions, growth cells actually favor glucose digestion to ATP-proficient mitochondrial oxidative phosphorylation [95], which is otherwise called the Warburg effect in PH. The reduction in glycolysis can also relieve vascular remodeling or cell proliferation. In mammals, the following four hexokinase (HK) isozymes have been identified: HK-I, HK-II, HK-III, and HK-IV. HK-II is a regulatory enzyme involved in energy metabolism and cell proliferation, with its two catalytic domains essential for its enzymatic activity. The enzyme catalyzes the conversion of glucose to glucose-6-phosphate with the participation of ATP and plays a pivotal role in regulating glycolytic flux [96]. In MCT-PH rats, HK-II expression and miRNA expression were increased. miR-125a-5p expression was found to be reduced by gene microarray and in PASMCs by RT-qPCR, and miR-125a-5p was found to contain a candidate binding site in the 3′-UTR of the HK-II mRNA through bioinformatic analysis of the targeting algorithms. Through this, the connection between miR-125a-5p and HK-II was examined. It was shown that miR-125a-5p decreased MCT-actuated glycolysis prompted by PASMCs by focusing on HK-II and hindering its proliferation to further develop MCT-PH [37].

### 3.3. miRNAs Involved in Ion Channels

Ion channels are specialized protein structures in cell membranes that allow the transmembrane transport of ions across cell membranes, and the permeability of membranes to cations (e.g., K^+^, Na^+^, and Ca^2+^) and anions (e.g., Cl^−^, HCO^2+^) plays an important role in the regulation of intracellular ionic homeostasis, cell volume, and excitability. Additionally, alterations in ion channel expression and function are key features in the development and pathogenesis of pulmonary vascular disease, particularly in PH, and Kv channel dysfunction and aberrant intracellular Ca^2+^ homeostasis are now widely recognized as important contributing factors in the pathogenesis of PH [97]. Recent studies have shown that miRNAs are also involved in ion channels, playing a regulatory role [38].

(1) miR-1: K channels are transmembrane proteins that associate intracellular and extracellular conditions by shaping pores in the cytoplasmic film. They are the biggest group of film particle channels, and K channels can be directed at various levels by factors that control channel expression, biosynthesis, specific chaperone protein transport, addition into the cell layer, reuse, debasement, and migration [98]. Additionally, the directions of K particles are associated with the pathogenesis of PH, and decreased K prevents migration in PASMCs, advancing cell proliferation, anti-apoptosis, and vasoconstriction and thus contributing to vascular remodeling [99]. Recent trial studies have shown that miR-1 changes in PH regulation affect Kv1.5 directions in rodent pulmonary corridors. The potassium voltage-gated channel subfamily A member 5 (kCNA5) mRNA in the Kv1.5 channel protein is the target of miR-1, and miR-1 directs Kv1.5 channels through this target. Among them, miR-1 diminishes the Kv1.5 current and prompts layer depolarization in PASMCs and antagomiR-1 was able to altogether expand Kv1.5 channel expression. These findings suggest that miR-1 reduces Kv channel migration and expression and plays a pathophysiological role in PH [38].

(2) miR-29b, miR-138, and miR-222: The resting layer potential (Em) in PASMCs controls the pulmonary blood vessel tone and vasoconstriction, and film depolarization causes PASMC compression and pulmonary vasoconstriction through the launch of voltage-subordinate Ca^2+^ channels (voltage-gated calcium channel, VDCC) in PASMCs, proliferating in ([Ca^2+^]cyt) and also leading to PASMC proliferation and migration [100]. Likewise, K^+^ channel action regulates Em and diminishes K^+^ flows by downregulating channel expression. Additionally, restraining channel action depolarizes PASMCs, opens the VDCC, and increases Ca^2+^ endocytosis [97]. It has been found that upregulation of miR-29b, miR-138, and miR-222 in IPAH-PASMC is accompanied by a decrease in KV channel activity. The inhibition of miR-29b preserved KV channel activity and expression in PASMC isolated from IPAH patients, and miR-29b regulated Ca^2+^ activated K^+^ channel (BKCa) activity and potassium calcium-activated channel subfamily M regulatory beta subunit 1 (KCNMB1) expression. These results suggest that miR-29b reduces the function and expression of KV and BKCa channel subunits and contributes to causing pulmonary vasoconstriction and pulmonary vascular remodeling. It is possible that miR-29b may play a therapeutic role in PH by regulating miR-29b [101]. Recent studies have shown that a reduced Kv current in PH ultimately leads to pulmonary vasoconstriction and induces PASMC proliferation and migration. miR-29b-2, a circular nuclear transcription factor, and X-box binding-like 1 (circNFXL1) sponge are involved in the regulation of Kv2.1 channels and mediate the hypoxia-induced increase in [Ca^2+^]cyt in PASMCs through membrane depolarization. However, the ways in which they affect vasoconstriction, PASMC proliferation, and migration remain unknown [39].

(3) miR-25 and miR-138: Vasoconstriction and cell proliferation in PAH are driven by elevated cytoplasmic calcium ([Ca^2+^]cyto), which occurs due to the increase in calcium transport through L-type calcium channels. The mitochondrial calcium uniporter (MCU) complex (MCUC) is a deep correcting Ca^2+^ specific particle located in the inward mitochondrial layer [102]. The regulation of PASMC migration, proliferation, and apoptosis was caused by MCUC inhibition, and miR-138- and miR-25-mediated post-transcriptional downregulation of MCU. Overall, downregulation of MCU by miR-138 and miR-25 increased the transcriptional regulator of MCU and cAMP-response element-binding protein 1 (CREB1), and the inhibition of miR-138 and miR-25 reversed the MCT rat model of PAH. The abovementioned findings suggest that miRNAs regulate ion channels by modulating MCU [40].

### 3.4. miRNAs Involved in Inflammation and Fibrosis

Fibrosis is a gradually advancing tissue response, degenerative in all cases, with destructive ramifications for heart, lung, liver, kidney, and skin illnesses [103]. Furthermore, irritation is a notable feature of PH, which is significant for the establishment and support of vascular remodeling [104]. Therefore, irritation and fibrosis play a critical role in PH. Previous studies have shown that a consistent decline in miR-124 expression contributes to the epigenetic reinvention of exceptionally proliferative, migration, and provocative aggregates of hypertensive lung external layer fibroblasts. The overexpression of miR-340-5p inhibits IL-1β and IL-6, which in turn inhibits the inflammation, cell proliferation, and migration of PASMCs, thereby inhibiting thrombosis and APE-induced PAH. Therefore, this indicates a role of miRNAs [81].

(1) miR-483: miR-483, including miR-483-3p and miR-483-5p, is a notable miRNA among those reported as involved in PAH and can target multiple PAH genes. Previous studies have shown that miR-483 promotes EC function, in part through its anti-fibrotic effects, and that an impaired flow reduces miR-483 levels in ECs [105]. In preliminary studies, reduced levels of miR-483 were identified in patients with IPAH. Subsequently, isolated CD144—a VE–calmodulin marker for endothelial cells—enriched extracellular vesicles (EVs) in the sera of IPAH patients and age- and sex-matched healthy controls (HCs). Analysis revealed that the miR-483-3p and miR-483-5p contents in CD144-enriched EVs derived from IPAH patients were significantly lower than those from HC subjects. These findings suggest that reduced miR-483 expression may influence the genes and pathways involved in PAH pathogenesis, potentially providing new insights into the mechanisms underlying this condition. Subsequent RNA-seq analysis demonstrated that miR-483 overexpression downregulates the genes involved in cell proliferation, inflammation, migration, apoptotic processes, responses to hypoxia and oxidative stress, and miR-483 targeting of IL-1β by the luciferase reporter. Furthermore, miR-483 has been shown to regulate inflammation in PH by targeting IL-1β. Subsequently, elevated levels of miR-483 were found in the epicardial mesenchyme of pulmonary arteries from EC-miR-483-Tg rats and in pulmonary arterial SMCs co-cultured with lung EC isolated from EC-miR-483-Tg rats. It was shown that miR-483 also inhibited SMC proliferation and inflammation [41]. The use of exogenous miR-483 or the enhancement of endogenous miR-483 could, therefore, offer a therapeutic approach for PH.

(2) miR-181a/b: The miR-181 family has been shown to be a key biomarker for a variety of diseases [106,107,108]. Previous studies have demonstrated that miR-181a/b inhibits lung injury by suppressing EC activation and reducing inflammation [109], but the role of miR-181a/b in PH has not been investigated. Endogenous glycans are key biomarkers of EC dysfunction and can influence the state of inflammation [110], and the upregulation of endoglycans in the PAH model is associated with the inflammatory response. It was observed that MCT infusion regulated miR-181a/b, endogenous glycans, and tumor necrosis factor-α (TNF-α). Overexpression of miR-181a/b was found to block MCT-actuated demise, increase hemodynamics, and cause right ventricular remodeling in rodents. Additionally, immunofluorescence staining showed that miR-181a/b weakened MCT-induced lung injury and expanded endogenous glycans and attachment particles in PAH rodents. In TNF-α-treated rPAECs, overexpression of miR-181a/b inhibited the increase in endoglycans in TNF-α-treated cells and also inhibited the action of TNF-α, together confirming that miR-181a/b plays a negative regulatory role in PAH by mitigating the inflammatory response through interaction with endogenous glycans [42]. However, miR-181a/b expression has not been studied in clinical samples, and there are many other target genes of miR-181a/b; its specific mechanisms in PH need to be further investigated.

(3) miR-429-3p: miR-429 plays a significant role in different organic cycles, including in the control of cancer, and miR-429 expression is associated with a limited endurance in osteosarcoma patients [111]. Moreover, miR-429 overexpression changed the ascorbic corrosive-prompted level of H9c2 cell practicality and lessened inflammatory injury [112], and downregulation of miR-429 in PAH restrained the proliferation of PASMCs, which then prompted a decline in Ca2+ efflux [113]. ITGB1 is an integrin subunit that is fundamental for extracellular network collaborations and numerous flagging channels that safeguard pulmonary veins through cell and cell–ECM communications [114]. Exosomes are small extracellular vesicles that are discharged by an assortment of cell types, including resistant cells, immature microorganisms, and growth cells. Exosomes play a key role in intercellular correspondence by contributing to the exchange of biomolecules between cells [115]. The level of miR-429-3p in the exosomes (ITGB1-Exo) of integrin subunit beta 1 (ITGB1)-modified TCs was significantly increased by ITGB1 overexpression. Furthermore, Rac-family small GTPase 1 (RAC1) was predicted to be a direct target gene of miR-429-3p by the luciferase reporter. Ectopic overexpression of Rac1 in Exo-ITGB1-treated PASMCs attenuated the inhibitory effect of Exo-ITGB1 on the cell viability of PASMCs, and the reduction in migratory capacity caused by Exo-ITGB1 was reversed, activating the production of inflammatory factors in Exo-ITGB1-treated PASMCs. This suggests that miR-429-3p regulates hypoxia-induced PASMC proliferation, migration, and inflammation by targeting Rac1 [43].

Many miRNAs that have a regulatory effect on PH inflammation and fibrosis; for example, miR-150 exerts anti-apoptotic, anti-inflammatory, anti-proliferative, and anti-fibrotic effects by decreasing the expression of inflammatory, apoptotic, and fibrotic markers that are critical for PAH pathology [116]. Levels of TNF-α, IL-1β, IL-18, and IL-6 were significantly elevated in CT-induced PASMCs from PAH rats overexpressing miR-15a-5p. In contrast, levels of TNF-α, IL-1β, IL-18, and IL-6 were reduced. This suggests that miR-15a-5p inhibits inflammation in PH [117]. The overexpression of miR-663b in PASMCs exacerbated their proliferation and migration, as well as inflammation and oxidative stress [118]. In conclusion, inflammation and fibrosis, as common features in PH, are involved in the regulation of many miRNAs, but most of them have already been studied as secondary targets.

### 3.5. miRNAs Involved in Right Ventricular Dysfunction

Right ventricular (RV) dysfunction includes right ventricular ischemia, right ventricular hypertrophy, right ventricular fibrosis, right ventricular metabolic abnormalities, and right ventricular hibernation. In recent decades, it has been increasingly recognized that right ventricular dysfunction plays a key role in the functional status and outcomes of a variety of diseases, such as PAH, congenital heart disease, and cardiomyopathy [119,120,121]. Emerging evidence indicates that many miRNAs are implicated in the regulation of right ventricular dysfunction in PH, positioning them as promising biomarkers for diagnosing right ventricular hypertrophy in pulmonary hypertension patients [122].

(1) miR-29a-3p: The miR-29 family plays a crucial role in multi-organ fibrosis. miR-29a-3p prevents schistosomiasis-induced activation of activated hepatic stellate cells (HSCs) during infection by targeting Robo1, thus partially reversing schistosome-induced hepatic fibrosis [123]. The inhibition of H19 alters miR-29a levels to inactivate the TGF-β/SMAD pathway, which downregulates the endothelial-to-mesenchymal transition (EndMT), leading to the inhibition of renal fibrosis [124]. THBS2 is a pro-fibrotic, anti-angiogenic stromal cell protein that is an important regulator of ECM homeostasis in the progression of fibrosis [125]. Increased ECM deposition and fibrosis in various tissues is associated with a decreased expression of miR-29 family members. Targeting miR-29a-3p to thrombospondin 2 (THBS2) in a mouse model of MCT-induced PAH and cardiac remodeling resulted in reduced cardiac fibroblast activation and activation of the pro-fibrotic pathway. In addition, the overexpression of miR-29a-3p inhibited cardiac fibroblast activation and fibroblast-to-myofibroblast transition, and significantly reduced the expression of pro-fibrotic markers [44].

(2) miR-335-5p: miR-335-5p is a tumor suppressor gene that inhibits tumor cell proliferation, migration, and invasion, and is involved in the regulation of cancers such as gastric cancer, breast cancer, and liver cancer [126,127,128]. Furthermore, miR-335-5p is associated with cardiomyocyte safety [129]. miR-335-5p was found to be upregulated in the right ventricle of PAH rodents. In an in vitro model of cardiac hypertrophy, miR-335-5p levels were also elevated, and miR-335-5p inhibition attenuated angiotensin II-induced cardiomyocyte hypertrophy. In addition, miR-335-5p was significantly reduced in R-335-5p-treated hypoxia/su5416 rats, and it attenuated hypoxia/su5416-induced right ventricular hypertrophy in PAH. The heart hypertrophy markers atrial natriuretic peptide (ANP) and myosin heavy chain beta (β-MHC) were also diminished after miR-335-5p inhibition, suggesting that miR-335-5p is associated with right ventricular remodeling in PAH, and that its inhibition attenuates this process [45].

(3) miR-21: miR-21 plays a critical role in cardiomyocyte proliferation, cell death, and hypertrophy [130]. In previous studies, miR-21 was found to be highly expressed in pulmonary artery tissues of several types of human PAH and rodent models and to be involved in vascular remodeling in a small animal model of PAH. miR-21 is a known enhancer of the pro-fibrotic intracellular pathway, and its upregulation has been associated with right ventricular fibrosis [131]. In a sheep PAH model, miR-21 expression was observed to be significantly increased by quantitative RT-qPCR, especially in the basal region of the right ventricle. By inhibiting miR-21 expression, cardiomyocyte hypertrophy could be reversed, thus acting in accordance with right ventricular remodeling [132]. In a different trial, in PAH rodents undergoing an adrenal vein sampling (AVS) medical procedure, a huge proliferation in miR-21 was noticed from the very beginning of life; however, this then diminished throughout life, linked with RV hypertrophy through the early compensatory ease and decompensated RV damage due to the consequent lifelong pulmonary hypertension. It was also tentatively confirmed that miR-21 was overwhelmingly upregulated in RV cardiomyocytes. Blood-stream-incited cardiomyocyte apoptosis was constricted by the overexpression of miR-21 to repress Spry2/PTEN and advance AKT/ERK phosphorylation. These findings indicate that the unique changes in miR-21 in PAH add to the biphasic regulation of heart remodeling and apoptosis. In all cases, only miR-21 was considered; other miRNAs might be involved and play a more important role in PAH with bloodstream over-burden [133]. The rise in miR-21 expression in induced RV fibroblasts also demonstrated the involvement of miR-21 in RV cardiomyocytes and RV fibroblasts, with both significantly increased. Attenuated proliferation of RV fibroblasts was found through the knockdown of miR-21 [134]. The abovementioned findings suggest that miR-21 is not only involved in right ventricular remodeling, but also in right ventricular fibrosis, and that inhibition of endogenous miR-21 or exogenous miR-21 antagonists may represent potential therapeutic targets for PH cardiac remodeling.

(4) miR-325-3p: Previous studies have shown that miR-325-3p inhibits renal inflammation and fibrosis by targeting C-C motif chemokine ligand 19 (CCL19) [135] and attenuates myocardial fibrosis after myocardial infarction by downregulating GLI-family zinc finger 1 (GLI1) [136]. Human epididymis protein 4 (HE4), a secreted protein expressed by activated fibroblasts, contributes to tissue fibrosis; as a secreted factor, HE4 activates myocardial fibroblasts and induces interstitial fibrosis [137]. Recent studies have shown that overexpression of miR-325-3p attenuates myocardial fibrosis in PAH rats and that HE4 is a target of miR-325-3p and its expression is negatively correlated. It was also confirmed experimentally that HE4 promotes cardiac fibroblast fibrosis through activation of the PI3K/AKT pathway. In conclusion, miR-325 can alleviate right ventricular fibrosis in PAH rats by targeting HE4 and regulating PI3K/AKT signaling [47].

In addition, many other miRNAs are involved in regulating right ventricular dysfunction in PH. For example, miR-646, miR-570, and miR-885 are sponged by CircRNA_0068481, an all-target EYA transcriptional coactivator, and phosphatase 3 (EYA3) mRNA. The suppression of miR-646, miR-750, and miR-885 and the upregulation of EYA3 mRNA expression are achieved through the upregulation of circRNA_0068481 expression, thereby promoting right ventricular hypertrophy in patients with PAH [123]. The expression of miR-663 (ad-miR-663) by an intratracheal drip significantly prevented MCT-induced right ventricular hypertrophy in vivo. This suggests that miR-663 may serve as a therapeutic target for PAH and that the identification of diagnostic biomarkers may allow for the early identification of patients at risk for PAH and provide new insights into the pathogenesis of the disease [138].

## 4. The Future of miRNAs in PH

### 4.1. miRNAs in PH as Diagnostic and Prognostic Biomarkers

Through their measurable and evaluative nature, biomarkers can serve as target indicators of normal biological phenomena, disease mechanisms, or drug responses [139]. Certain biomarkers should be utilized to distinguish illnesses, as should unambiguous endotypes/aggregates that effectively visualize negligible distresses or hazards. Previous studies have shown that miRNAs are consistently present and stable in serum samples, allowing for subsequent exploration of miRNAs as biomarkers for various diseases [140]. However, to date, brain natriuretic peptide (BNP) and NT-proBNP remain the only biomarkers that are widely used in routine practice and clinical trials in PH centers [141]. Right heart catheterization, despite being the gold standard for diagnosing PH, is unacceptably invasive. The search for better biomarkers is, therefore, crucial. Serum miR-21 has been reported to be upregulated in children with CHD-PAH, and miR-21 has been identified as an independent risk factor for the development of PAH in children with CHD. Animal experiments showed that early miR-21 upregulation (right ventricular hypertrophy) and late miR-21 downregulation (right ventricular dysfunction) triggered biphasic regulation of cardiac remodeling and cardiomyocyte apoptosis in pulmonary hypertension [133]. These findings suggest miR-21 upregulation and may serve as a predictive biomarker for PAH onset and postoperative critical illness in children with CHD [142]. In addition, another study found that the expression of hsa-miR-21-3p and hsa-miR-143-3p was significantly reduced in the plasma of PH patients and healthy individuals and that hsa-miR-143-3p, in particular, could be considered as a potential biomarker in the early diagnosis of PAH [23]. In animal experiments, differentially expressed miRNAs between PAH rats and LPS-treated PAH rats were identified by studying microarray analysis, and it was found that postoperative plasma levels of miR-212a-3p were lower in PAH rats after surgery. Thus, the stable expression of miR-212a-3p in plasma may be considered as a new early diagnostic marker [143]. Although miRNAs have become routine biomarkers in PH, they are not specific. For example, miR-30d, miR-124, and miR-150 are all downregulated in PH, but this is also the case in other diseases [144,145,146]. However, as a prognostic biomarker, it would be reasonable to only use the values of miRNAs before and after observing the status of PH patients’ recovery. In addition, there is a lack of longitudinal or large sample size studies, and the samples taken in most experiments are too small to be generally representative.

### 4.2. miRNAs as Therapeutic Agents for PH

Many previous experiments were performed by overexpressing or using miRNA inhibitors to modulate miRNAs, and from this, novel therapeutic agents for PH can be developed. miRNAs are easy to obtain, and as one miRNA can target multiple genes, its therapeutic pathways are diversified. Currently, miRNA mimics, miRNA antagonists, and EVs are the main ways in which miRNAs are used as therapeutic agents with the aim of enhancing or inhibiting the amount of endogenous miRNA or treating PH through exogenous miRNA (Table 2) Based on the databases clinicaltrialsregister.eu and clinicaltrials.gov, miRNA drugs used in clinical trials were compiled.

EVs are derived from mesenchymal stem cells (MSCs), a new axis of intercellular communication that binds to cell surface receptors and secretes its contents across the plasma membrane into target cells, allowing interaction with them. EVs’ properties underscore their potential as a promising therapeutic tool for cardiac tissue regeneration. Cardiomyocytes, endothelial cells, fibroblasts, and stem cells release EVs, which play a critical role in both pathological and physiological processes, including cardiac hypertrophy, cardiomyocyte survival and apoptosis, cardiac fibrosis, and angiogenesis associated with cardiovascular disease [147]. There is increasing evidence that EVs have important regulatory properties for cardiovascular biology and disease [148]. Exosomes are a class of small extracellular vesicles (30–150 nm in diameter) that are derived from the internal vesicles of multivesicular bodies and are universally present in various cell types. These vesicles contain a diverse array of non-coding RNAs, including microRNAs, long-stranded non-coding RNAs (lncRNAs), tRNA fragments, small interfering RNAs, structural RNAs, small RNA transcripts, and RNA–protein complexes (Figure 4) [149]. Bone marrow mesenchymal stem cell-derived extracellular vesicles (BMSC-EVs) have reportedly been used in the treatment of PH, and researchers have found that BMSC-EVs containing miR-200b can be internalized by alveolar macrophages and can attenuate PAH by promoting M2 macrophage polarization via the PDE1A/PKA axis [150]. Another study used telangiectasia cell (TC)-derived exosomes modified by ITGB1 and enriched with miR-429-3p. It was shown that miR-429-3p regulates the hypoxia-induced proliferation, migration, and inflammation of PASMCs by targeting Rac1. Moreover, Exo-ITGB1 treatment inhibited pulmonary vascular remodeling, reduced inflammatory cytokine production, and slowed down PAH progression in a hypoxia-induced common PAH model [43]. Further studies on the mechanism of action and its efficacy are needed before it can be promoted as a therapeutic agent for PH stabilization.

The application of miRNA therapy in PH is still in the preclinical stage; stability, delivery, and off-target effects remain the three main issues limiting its use [151]. Firstly, the RNA molecule has a 2′-OH chemical group and is therefore very unstable [152]. Secondly, the delivery of miRNA to the desired site of action remains an obstacle; it is important to ensure delivery to the target site while maintaining therapeutic specificity. To address this problem, recent studies have shown extracellular vesicles, nebulization, and nanocarriers to be possible solutions, but further clinical studies are required to ensure that they are more stable and safer as a therapeutic [153]. Finally, the off-target effects of miRNAs mainly disrupt cellular pathways unrelated to disease pathogenesis and induce immune responses, leading to toxic effects. Therefore, improving the specific delivery of miRNAs is key to reducing their off-target effects. Whether as biomarkers or therapeutic agents, further experimental studies and clinical trials with larger cohorts are needed to evaluate these miRNAs.

## 5. Conclusions

Although miRNAs are short RNA molecules that are only 19 to 25 nucleotides in size, they play a hugely significant role in various diseases. miRNAs are involved in every pathogenesis pathway for PH and play a potential role in them. PH has been associated with miRNAs from diagnosis to treatment, and miRNAs have subsequently been developed as biomarkers and therapeutic agents for PH. In PH risk stratification, miRNA can serve as a new standard and can also be used as a treatment strategy for each stratification. While miRNAs have potential regulatory roles in vascular remodeling, inflammation, and fibrosis, the currently available studies have focused on demonstrating indirect PH regulation by various miRNAs, a single miRNA can have hundreds of target sites and participate in multiple signaling pathways, making the mechanism too complex. The lack of clinical trials and individual heterogeneity are the main reasons why miRNAs have not yet been introduced into clinical practice, and they are also the limitations of miRNAs in PH research. To become good biomarkers and therapeutic agents, more clinical studies are needed.

## Figures and Tables

**Figure 1 biomedicines-13-01910-f001:**
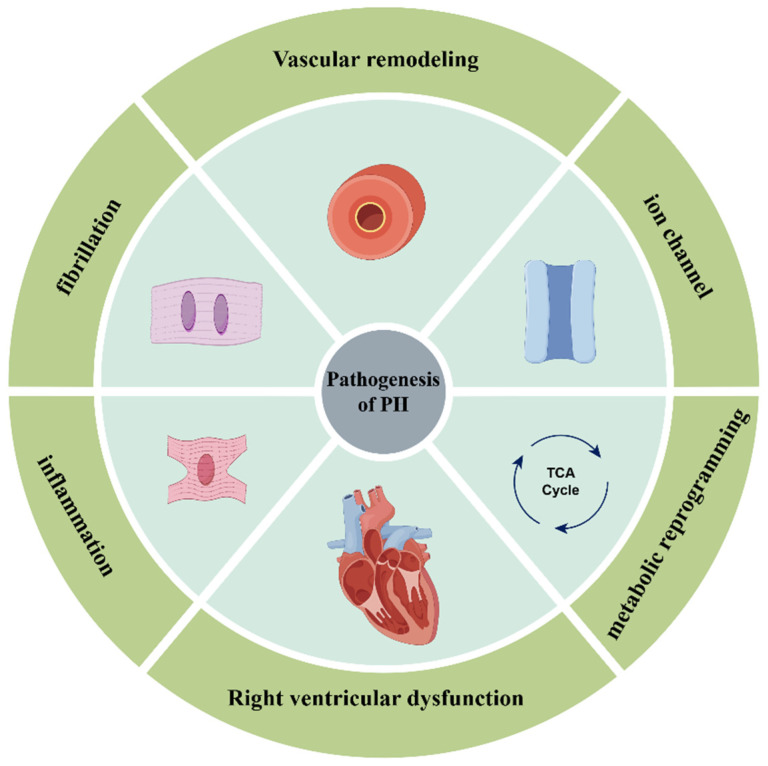
The main pathogenesis of pulmonary hypertension. The major mechanisms are vascular remodeling, metabolic reprogramming, right ventricular dysfunction, ion channels, inflammation, and fibrosis.

**Figure 2 biomedicines-13-01910-f002:**
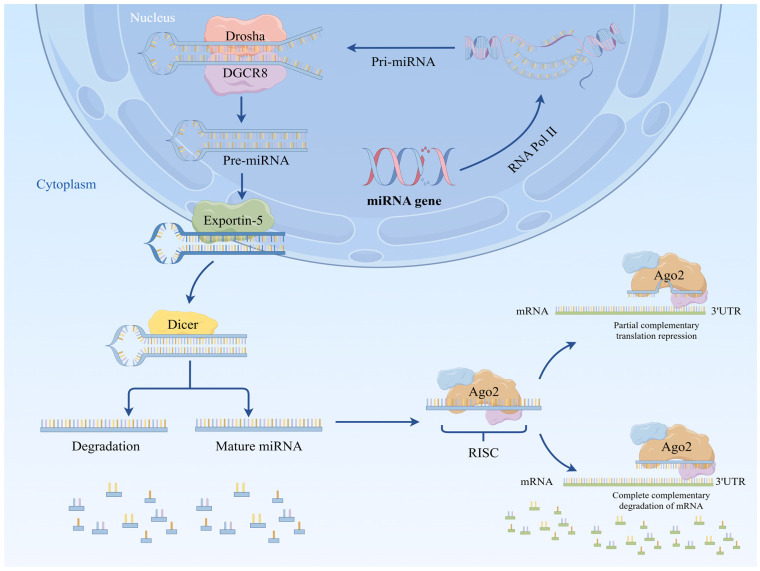
The miRNA biosynthetic pathway, including transcription of genes, processing of primary transcripts and production of mature miRNAs.

**Figure 3 biomedicines-13-01910-f003:**
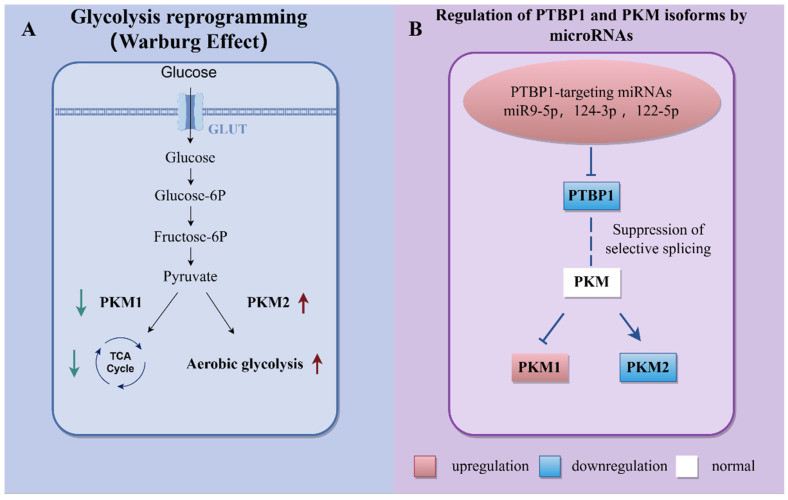
Regulation of metabolic reprogramming through microRNA regulation of PTBP1 and pyruvate kinase M (PKM) isoforms. (**A**). Warburg effect. (**B**). Regulation of polypyrimidine tract-binding protein 1 (PTBP1) and pyruvate kinase M (PKM) isoforms by microRNAs: the upregulation of miRNAs leads to the downregulation of PTBP, which in turn induces the upregulation of PKM1 and the downregulation of PKM2 through suppression of selective splicing, thereby reversing glycolysis.

**Figure 4 biomedicines-13-01910-f004:**
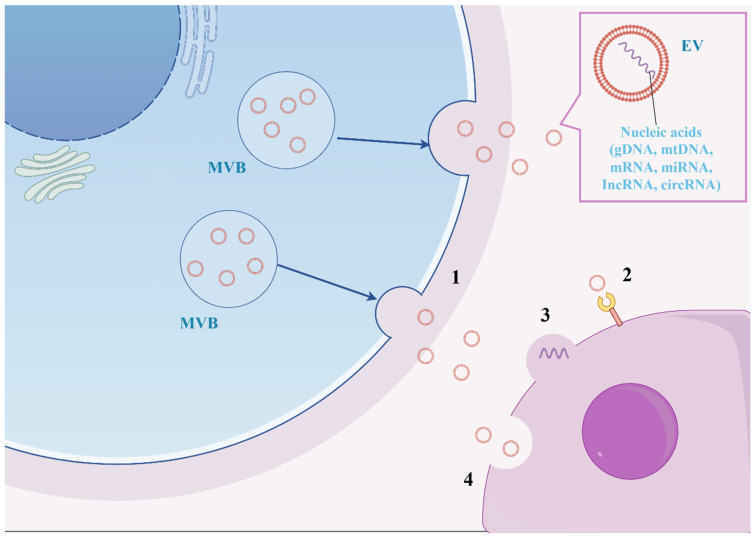
Extracellular vesicles (EVs) involved in the exchange of information between cells: (1) release of exosomes through fusion of MVBs with the plasma membrane; (2) vesicle binding to extracellular receptors; (3) vesicle fusion with plasma membrane; (4) vesicle uptake by endocytosis.

**Table 1 biomedicines-13-01910-t001:** miRNAs and downstream targets in the pathogenesis of pulmonary hypertension.

miRNA	Target	Disease	Reference(s)
miR-30d	MTDH PDE5A	Pulmonary hypertension	[24]
miR-212-5p	Unknown	Pulmonary hypertension	[25]
miR-340-5p	MFF	Pulmonary hypertension	[26]
miR-153	ROCK1 NFATc3	Pulmonary hypertension	[27]
miR-27	PPARγ	Pulmonary hypertension	[28,29,30]
miR-30a-5p	YKL-40	Pulmonary hypertension	[31]
miR-410	NAMPT	Pulmonary hypertension	[32]
miR-124	PTBP1	Pulmonary hypertension Idiopathic pulmonary arterial hypertension	[33,34]
miR-22-3p	C10:2	Pulmonary hypertension	[35]
miR-329-3p	PDHB	Pulmonary hypertension	[36]
miR-125a-5p	HK-II	Pulmonary hypertension	[37]
miR-1	Kv1.5 channel	Pulmonary hypertension	[38]
miR-29b, miR-138 and miR-222	Kv2.1 channel	Pulmonary hypertension	[39]
miR-25 and miR-138	MCU	Pulmonary hypertension	[40]
miR-483	TGF-β TGFBR2 β-catenin CTGF IL-1β ET-1	Pulmonary hypertension	[41]
miR-181a/b	Endocan	Pulmonary hypertension	[42]
miR-429-3p	Rac1	Pulmonary hypertension	[43]
miR-29a-3p	THBS2	Pulmonary hypertension	[44]
miR-335-5p	CALU	Pulmonary hypertension	[45]
miR-21	FilGAP	Pulmonary hypertension	[46]
miR-325-3p	HE4	Pulmonary hypertension	[47]

**Table 2 biomedicines-13-01910-t002:** miRNA drugs used in clinical trials.

miRNA Drug Name	Targeted miRNA	Disease/Condition	Clinical Phase
INT-1B3	miR-193a-3p	Advanced solid tumors	Phase I
RC.012/lademirsen/SAR339375	miR-21	Alport syndrome	Phase II
RGLS4326	miR-17	Autosomal dominant polycystic kidney disease	Phase I
RG-125/AZD4076	miR-103/107	T2DM with NAFLD	Phase I
Remlarsen/MRG201	miR-29	Keloid disorder	Phase II
Miravirsen/SPC3649	miR-122	Chronic hepatitis Cvirus	Phase II
MRX34	miR-34a	Primary liver cancer, SCLC, lymphoma, multiple myeloma, renal cell carcinoma, NSCLC	Phase I
Cobomarsen/MRG-106	miR-155	Mycosis fungoides (MF), cutaneous T-cell lymphoma (CTCL), chronic lymphocytic leukemia (CLL), diffuse large B-cell lymphoma (DLBCL), ABC subtype adult T-cell leukemia/lymphoma (ATLL)	Phase II

## Data Availability

No new data were created or analyzed in this study. Data sharing is not applicable to this article.

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
