# Peer review of "miRNAs in Pulmonary Hypertension: Mechanistic Insights and Therapeutic Potential"

_biomedicines, 2025, doi:10.3390/biomedicines13081910_

Round 1
Reviewer 1 Report
Comments and Suggestions for Authors
This review manuscript presents an overview of the role of miRNAs in the pathogenesis and treatment of pulmonary hypertension. The authors summarize current evidence on miRNA involvement in various pathological processes, including vascular remodeling, metabolic reprogramming, inflammation, ion channel dysregulation, and right ventricular dysfunction. However, several critical issues require revision.
1. The manuscript contains numerous grammatical errors, awkward phrasing, and inconsistent terminology (e.g., "aspiratory" instead of "pulmonary", "expansion" instead of "proliferation").
2. Some important miRNAs (e.g., miR-1298, miR-96-5p) are missing. recommend author to cite PMID: 39160536 and PMID: 37344798
3. Line 28: mPAP should be "mean pulmonary arterial pressure," not "mean aspiratory vein".
4. Consider reformatting long paragraphs (e.g., Section 3.1) for better readability.
5. In Section 3.1 which discusses the role of miRNAs in vascular remodeling, the term "hypoxia-induced PH rodent model" is mentioned multiple times. The latest reference, PMID: 38322991, should be cited here.
Author Response
Response to Reviewer 1 Comments 1.The manuscript contains numerous grammatical errors, awkward phrasing, and inconsistent terminology (e.g., "aspiratory" instead of "pulmonary", "expansion" instead of "proliferation"). Response 1: Thank you very much for your professional comments and suggestions on this review. Accurate grammar, terminology and jargon are essential for correctly expressing the content of the text. We fully agree with your comments and have corrected the problems that appeared in the review as requested: We have changed aspiratory to pulmonary, expansion to proliferation, and standardised other terminology. Thank you again for your professional comments. In addition to this, we have also performed the professional English editing through MDPI’s Author Services throughout the review, which has enhanced the standardisation, logic and dissemination of this review. 2.Some important miRNAs (e.g., miR-1298, miR-96-5p) are missing. recommend author to cite PMID: 39160536 and PMID: 37344798 Response 2: Thank you very much for your valuable comments. Important research reports are better able to reflect the latest advances and give inspiration to researchers to advance their scientific endeavours. Based on your comments I have added these two important papers to the review and elaborated on them. The modifications are as follows: L 244-249, Activation of the BMP/Smad signaling pathway through the downregulation of miR-1298 inhibits cell proliferation and migration and promotes apoptosis in hypoxia-treated PASMCs [48]. CircItgb5 promoted the transition of PASMCs to a synthetic phenotype by interacting with miR-96-5p and ubiquitin-like modifier-activating enzyme 1 (Uba1), further regulating pulmonary vascular remodeling [49]. These processes are involved in smooth muscle remodeling in PH vascular remodeling. 3.Line 28: mPAP should be "mean pulmonary arterial pressure," not "mean aspiratory vein". Response 3: Thank you very much for your valuable suggestions. Accuracy in professional names is necessary, as per your suggestion the following changes have been made: L 29, “mean aspiratory vein corrected to pulmonary arterial pressure”. and again checked the proper names appearing in the review, thanks again for your suggestion! 4.Consider reformatting long paragraphs (e.g., Section 3.1) for better readability. Response 4: Thank you for your valuable advice. Improving the readability of the review will better help the readers to understand the content of the review. Based on your suggestion I have optimised the content of long sentences and made sentence breaks. It is as follows: L 158-171, “Due the variation in the increase in certain factors, including in hypoxia, irritation, and oxidative pressure, the pulmonary arteriovenous framework changes, prompting pulmonary vascular remodeling and eventually advancing PH [24]. Additionally, during pulmonary vascular remodeling in patients with PH, vascular endothelial injury, vascular media hypertrophy, fringe vascular muscle fibrosis, and an expanded extracellular matrix (ECM) are often observed. Thus, the pulmonary vascular lumen becomes contracted, pulmonary conduits become blocked, and vasoproliferative plexiform injuries occur, which cause PH to develop [25]. Pulmonary vascular remodeling is a vital element of PH, which includes the proliferation of and phenotypic changes in intimal pulmonary arterial endothelial cells (PAECs) and pulmonary artery smooth muscle cells (PASMCs), as well as complicated connections including, among others, the external layer of pulmonary arterial fibroblasts (PAFs). In ongoing studies, miRNAs have been shown to be communicated in these cell types; they play a significant role in determining the aggregate of every cell type and in vascular remodeling components [26].” Thanks again for your comments! 5.In Section 3.1 which discusses the role of miRNAs in vascular remodeling, the term "hypoxia-induced PH rodent model" is mentioned multiple times. The latest reference, PMID: 38322991, should be cited here. Response 5: Thank you very much for your professional opinion. The latest literature better reflects the progress of research and gives normal guidance to the researcher. As per your suggestion I have added the literature to the review where the model first appears. Below: L 193,[31]
Reviewer 2 Report
Comments and Suggestions for Authors
This review manuscript shows significant promise and provides essential insights into the impact of miRNAs in pulmonary hypertension, but it requires substantial revisions before publication in this journal.
- What does ESC stand for? Provide a full form of all abbreviations at first sight.
- Line30-40, in the ESC (European Society of Cardiology) guidelines or emergency room (ER) protocols, often used to prioritize treatment based on urgency. However, your phrase “gathering 1, gathering 2, and gathering ?” is unclear — you may be referring to triage levels, clinical groupings, or risk stratification classes.
- Cite references from lines 24-35. Only one reference cited….
- Line 50 showed that…. “halt the progression of PH by targeting the underlying molecular mechanisms” mention mechanisms like?????. Also references are needed from lines 51-53.
- On line 54 “gander”???? explain….
- References are missing lines 62 to 82,93-106. 142 need reference
- Line 164,181, what does it mean, “Then, I will portray the…..”??/ “I will introduce the administrative jobs ”…???
- Line 191, the first letter of Advantage ,Cardiovascular, Remodeling should be small.
- Line 205,224, As of late mean???? And need to cite references at the end of each sentences 205-214 and make sentences shorter for easy to understand.
- I suggest authrs, instead of "as of late," consider more formal and precise phrases such as:"Recently", "In recent studies", "In recent years", "Lately" (still slightly informal, use with caution), "Over the past decade" (if time frame is longer and defined)
- CT-1 on line 220,stand for???
- Rewrite sentence from 234-237.
- I did not hear before these technique names are newly developed???.., “western smudge investigation, and cell relocation capacity was not entirely settled by wound mending and Transwell measure” please go to reference articles and write the correct technique names.
- Cite references at 259.
- Line 261, Phosphatase And Tensin Homolog, write small first letters.
- Line 296, correct it “Be that as it may, its job in PAH is muddled”
- Need to cite proper references from 295-306.
- Rewrite this sentence, break it into short sentences from 332-338, and cite references at the end of each sentence.
- 451-458 very long sentences.
- References cited 461-465.
- 555 to 557 references are needed.
- Overall, Significant adjustments are necessary, especially concerning citations and clarity.
- The lack of references—ensure that all methodologies are correctly cited.
- It requires comprehensive revisions to interpret the results better, integrate current literature, and emphasize the broader implications of the findings.
Author Response
Response to Reviewer 2 Comments
1.What does ESC stand for? Provide a full form of all abbreviations at first sight.
Response 1: We are very grateful for your professional comments on the language presentation of this review. Correctness of language is the basis of the review, and I have revised it according to your comments. It is as follows: L 30-31, ESC (European Society of Cardiology) guidelines or emergency room (ER) protocols. We thank you again for your rigorous monitoring of the linguistic standardization. In addition to this, we have also performed the professional English editing through MDPI’s Author Services throughout the review, which has enhanced the standardisation, logic and dissemination of this review.
2.Line30-40, in the ESC (European Society of Cardiology) guidelines or emergency room (ER) protocols, often used to prioritize treatment based on urgency. However, your phrase “gathering 1, gathering 2, and gathering ?” is unclear — you may be referring to triage levels, clinical groupings, or risk stratification classes.
Response 2: We sincerely thank you for taking the time out of your busy schedule to review my comments. Based on your comments, I reviewed the relevant literature and made the correct formulation and have replaced gathering with clinical group (L 31-34). We thank you again for your rigorous monitoring of the linguistic standardization.
3.Cite references from lines 24-35. Only one reference cited….、
Response 3: Thank you very much for your professional comments and suggestions on our work. Following your suggestion, we have added a specialised literature to support the language. It is as follows: L 30,[1] 1.Mocumbi, A., et al., Pulmonary hypertension. Nat Rev Dis Primers, 2024. 10(1): p. 1. We thank you again for your rigorous monitoring of the linguistic standardization.
4.Line 50 showed that…. “halt the progression of PH by targeting the underlying molecular mechanisms” mention mechanisms like?????. Also references are needed from lines 51-53.
Response 4: We are very grateful for your professional comments on the language presentation of this review. We have listed specific mechanisms based on your suggestions and added references for L 51-53. as follows: L 52-55, In addition, therapeutic modulation of miRNAs has emerged as a promising intervention strategy with the potential to reverse or halt the progression of PH by targeting the underlying molecular mechanisms. One method is to intervene in the processing of RNAs by targeting, for example, pri-mRNAs and pre-miRNAs [6]. We thank you again for your rigorous monitoring of the linguistic standardization.
5.On line 54 “gander”???? explain….
Response 5: We sincerely thank you for taking the time out of your busy schedule to review my comments. Sorry for the appearance of non-scientific language. I have amended the original sentence. It is as follows: L55-57, This review explores the pathogenesis of PH and describes the associated miRNAs. In addition, future directions for miRNA therapy are discussed. We thank you again for your rigorous monitoring of the linguistic standardization.
6.References are missing lines 62 to 82,93-106. 142 need reference
Response 6: We are very grateful for your professional comments on the language presentation of this review. Based on your comments, references have been added to the review at the appropriate places. They are as follows: L 63-83, L 93-104, L 136. We thank you again for your rigorous monitoring of the linguistic standardization.
7.Line 164,181, what does it mean, “Then, I will portray the…..”??/ “I will introduce the administrative jobs ”…???
Response 7: We sincerely thank you for taking the time out of your busy schedule to review my comments. Scientific presentation is critical to the review, and I have removed the non-scientific language you pointed out. We thank you again for your rigorous monitoring of the linguistic standardization.
8.Line 191, the first letter of Advantage ,Cardiovascular, Remodeling should be small.
Response 8: We are honored and excited to have your review and guidance. The following changes have been made in response to your suggestions: L 180-181 advantages of miR-30d in cardiovascular remodeling. We thank you again for your rigorous monitoring of the linguistic standardization.
9.Line 205,224, As of late mean???? And need to cite references at the end of each sentences 205-214 and make sentences shorter for easy to understand.
Response 9: We are very grateful for your professional comments on the language presentation of this review. Because of your professional opinion, make our review more rigorous, more scientific, first we have made the following changes according to your requirements: L 192-201. In recent studies in hypoxic rats, miR-212-5p was found to be upregulated in pulmonary vascular cells. Similarly, in hypoxia-induced PH rats [31], this microRNA was also upregulated in the lungs. Based on these findings, a series of experimental studies were carried out to explore the role of miR-212-5p in this context. Firstly, miR-212-5p was upregulated in PASMCs and in the lungs of PH patients and rodents, as assessed by qPCR examination. Following this observation, in vitro cell tests utilizing the miR-212-5p antagonist (Anti-miR-212-5p) to decipher the role of miR-212-5p in PASMCs and PH revealed that miR-212-5p is an anti-proliferative miRNA in PASMCs. Endogenous miR-212-5p was also shown to have a defensive effect on SMCs with PH by SMC-specific knockout of miR-212-5p mouse strains [32]. We thank you again for your rigorous monitoring of the linguistic standardization.
10.I suggest authrs, instead of "as of late," consider more formal and precise phrases such as:"Recently", "In recent studies", "In recent years", "Lately" (still slightly informal, use with caution), "Over the past decade" (if time frame is longer and defined)
Response 10: We are very grateful for your professional comments on the language presentation of this review. Phrases have been changed based on your comments. As follows: L 192, In recent studies. We thank you again for your rigorous monitoring of the linguistic standardization.
11.CT-1 on line 220, stand for???
Response 11: We are honored and excited to have your review and guidance. I apologise for our lack of clarity in this section, which we have elaborated on in order to enhance the readability of the review. As follows: L 204-205, Cardiotrophin-1 (CT-1). We thank you again for your rigorous monitoring of the linguistic standardization.
12.Rewrite sentence from 234-237.
Response 12: We are very grateful for your professional comments on the language presentation of this review. I apologise for our lack of clarity in this section, which we have elaborated on in order to enhance the readability of the review. As follows: L 216-221, miR-153 is chiefly engaged in various pathophysiological cycles of human illnesses such as hindrance of cellular breakdown in the lungs [39], restraint of pulmonary fibrosis [40], and hindrance of angiogenesis under hypoxic conditions [41], in addition to having an impact on PAH. Recent studies have shown that pulmonary vascular remodeling under hypoxic conditions is mainly caused by the proliferation and migration of human pulmonary artery smooth muscle cells (HPASMCs) [42]. Thanks again for contributing to the science of the review! In addition to this, we have also performed the professional English editing through MDPI’s Author Services throughout the review, which has enhanced the standardisation, logic and dissemination of this review.
13.I did not hear before these technique names are newly developed???.., “western smudge investigation, and cell relocation capacity was not entirely settled by wound mending and Transwell measure” please go to reference reviews and write the correct technique names.
Response 13: We are honored and excited to have your review and guidance. As per your suggestion, the error in the terminology has been corrected and red-flagged in the text. As follows: L 225-226, Western blot analysis, wound healing (cell migration) assays, and Transwell cell migration assays. We thank you again for your rigorous monitoring of the linguistic standardization.
14.Cite references at 259.
Response 14: We are very grateful for your professional comments on the language presentation of this review. Based on your suggestions, the references have been adjusted to ensure that they are located in the proper place. As follows: L 235-237, this only shows that miR-153 targets ROCK1 and NFATc3 to some degree, and investigating the specific mechanism of miR-153 in PH remains necessary [43]. Thank you again for your professional comments.
15.Line 261, Phosphatase And Tensin Homolog, write small first letters.
Response 15: We are deeply grateful for your hard work and patience in reviewing our work. The phrase has been written correctly based on your comments. As follows: L 239, phosphatase and tensin homolog. We thank you again for your rigorous monitoring of the linguistic standardization.
16.Line 296, correct it “Be that as it may, its job in PAH is muddled”
Response 16: We are very grateful for your professional comments on the language presentation of this review. Based on your suggestion, we have removed this type of non-scientific language to ensure the scientific nature of the review. We thank you again for your rigorous monitoring of the linguistic standardization.
17.Need to cite proper references from 295-306.
Response 17: We are deeply grateful for your hard work and patience in reviewing our work. Following your suggestion, we have added references to the paragraph to make it more linguistically rigorous. As follows: L 274-275, miR-30a-5p is involved in cell proliferation and apoptosis [57]; Thank you again for your professional comments.
18.Rewrite this sentence, break it into short sentences from 332-338, and cite references at the end of each sentence.
Response 18: We are very grateful for your professional comments on the language presentation of this review. Following your suggestion, we have split the paragraph into three short sentences and added references at the end of the sentences. As follows: L 306-312, Furthermore, miRNA-31-5p is involved in spermine-induced autophagy by targeting N-acetyltransferase 8-like (NAT8L) to regulate PAECs [64]. Platelet-derived growth factor (PDGF) affects miR-409-5p expression and regulates pulmonary artery endothelial cell dysfunction [65]. In addition, overexpression of miR-150-5p attenuates ox-LDL-induced endothelial cell injury in human venous endothelial cells (HUVECs) [66], and is also involved in the regulation of endothelial cells in PH, indicating the importance of endothelial cell regulation in PH vascular remodeling. We thank you again for your rigorous monitoring of the linguistic standardization.
19.451-458 very long sentences.
Response 19: We are very grateful for your professional comments on the language presentation of this review. Based on your comments, we have broken the long sentence and optimised the sentence content to make it more rigorous. As follows: L 421-429, Ion channels are specialized protein structures in cell membranes that allow the transmembrane transport of ions across cell membranes, and the permeability of membranes to cations (e.g., K+, Na+, and Ca2+) and anions (e.g., Cl-, HCO2+) plays an important role in the regulation of intracellular ionic homeostasis, cell volume, and excitability. Additionally, alterations in ion channel expression and function are key features in the development and pathogenesis of pulmonary vascular disease, particularly in PH, and Kv channel dysfunction and aberrant intracellular Ca2+ homeostasis are now widely recognized as important contributing factors in the pathogenesis of PH [89]. Recent studies have shown that miRNAs are also involved in ion channels, playing a regulatory role [90]. We thank you again for your rigorous monitoring of the linguistic standardization.
20.References cited 461-465.
Response 20: We are deeply grateful for your hard work and patience in reviewing our work. Following your suggestion, we have added a reference to the paragraph. As follows: L 430-434, K channels are transmembrane proteins that associate intracellular and extracellular conditions by shaping pores in the cytoplasmic film. They are the biggest group of film preview channels, and K channels can be directed at various levels by factors that con-trol channel expression, biosynthesis, specific chaperone protein transport, addition into the cell layer, reuse, debasement, and migration [91]. Thank you again for your professional comments.
21.555 to 557 references are needed.
Response 21: We are very grateful for your professional comments on the language presentation of this review. Following your suggestion, we have added a reference to the paragraph. As follows: L 512-514, but the role of miR-181a/b in PH has not been investigated. Endogenous glycans are key biomarkers of EC dysfunction and can influence the state of inflammation [105], Thank you again for your professional comments.
22.Overall, Significant adjustments are necessary, especially concerning citations and clarity.
Response 22: We are deeply grateful for your hard work and patience in reviewing our work. We have accepted the English editing services provided by MDPI to systematically revise the language throughout the text, to standardize terminology and to ensure the scientific validity of the scholarly presentation, as we are not native speakers of English. Your valuable suggestions have greatly contributed to the refinement of the academic expression in this paper, and we thank you again for your rigorous monitoring of the linguistic standardization.
23.The lack of references—ensure that all methodologies are correctly cited.
Response 23: Thank you very much for your advice in response to the missing references within the review and the references being in the wrong place. We have carried out a thorough check of the review to ensure that the references are complete and in the correct place. Thank you again for your professional comments.
24.It requires comprehensive revisions to interpret the results better, integrate current literature, and emphasize the broader implications of the findings.
Response 24: We are very grateful for your professional comments on the language presentation of this review. We have revised the full text. First, unreasonable language has been adjusted. Secondly, the content of the review was reconfirmed with references to ensure that the results were clearly presented. Finally, we have also performed the professional English editing through MDPI’s Author Services throughout the review. Your comments make the review more scientific as well as readable and help the scientific reader to fully understand the current state of research. Thank you again for your comments!
Reviewer 3 Report
Comments and Suggestions for Authors
The article submitted for review is undoubtedly relevant. It is written in a logical and well-structured manner. However, there are a few suggestions:
After describing all the microRNAs, provide a summary table listing all the discussed microRNAs, their respective targets, the diseases they are associated with, and how each microRNA influences the expression of specific genes.
Author Response
Response to Reviewer 3 Comments
After describing all the microRNAs, provide a summary table listing all the discussed microRNAs, their respective targets, the diseases they are associated with, and how each microRNA influences the expression of specific genes.
Response 1: Thank you for this valuable suggestion. We fully agree that a consolidated summary table will enhance the clarity and utility of the review. As requested, a comprehensive summary table (Table 1) has been added in the Section 3“ miRNAs involved in the pathogenesis of PH”, synthesizing all discussed microRNAs along with their targets, disease associations, and gene regulatory roles in pulmonary hypertension pathogenesis. Additionally, the manuscript has undergone professional English editing through MDPI’s Author Services, further enhancing its scientific clarity and readability.

Round 2
Reviewer 2 Report
Comments and Suggestions for Authors
The authors have addressed my concerns, and the manuscript should be accepted after the revisions are made.
1. Authors have discussed the diagnosis and prognosis of PH, but need to discuss risk stratifications.
2. Also, discuss the limitations of the miRNA studies in PH.
3. Cite references in Table 2, same as table1.
4. Provide high-resolution figures 1and 3.
Author Response
Response to Reviewer 2 Comments
1.Authors have discussed the diagnosis and prognosis of PH, but need to discuss risk stratifications.
Response 1: Thank you very much for your professional comments and suggestions on our work. The completeness of the content ensures the comprehensiveness of the review, and we have supplemented this part in the review. As follows: L 734-735, In PH risk stratification, miRNA can serve as a new standard and can also be used as a treatment strategy for each stratification. Thank you again for your professional evaluation of the review.
- Also, discuss the limitations of the miRNA studies in PH.
Response 2: We are deeply grateful for your hard work and patience in reviewing our work. Based on your suggestion, we have included the corresponding discussion in the review. As follows: L 738-742,a single miRNA can have hundreds of target sites and participate in multiple signaling pathways, making the mechanism too complex. The lack of clinical trials and individual heterogeneity are the main reasons why miRNAs have not yet been introduced into clinical practice, and they are also the limitations of miRNAs in PH research. Thank you again for your comments. Your comments have made the article more scientific and rigorous.
- Cite references in Table 2, same as table1.
Response 3: We sincerely thank you for taking the time out of your busy schedule to review my comments. Accurate data sources can enhance the scientific rigor and reliability of the review. Our data is sourced from two databases, and we have cited these sources at appropriate locations throughout the review. As follows: L 682-683, Based on the databases clinicaltrialsregister.eu and clinicaltrials.gov, miRNA drugs used in clinical trials were compiled. Thank you again for your professional comments.
4.Provide high-resolution figures 1and 3.
Response 4: We are deeply grateful for your hard work and patience in reviewing our work. High-resolution figures can enhance the scientific rigor and readability of the review. We have prepared high-resolution figures and included them in the attachments for your convenience. Thank you again for your professional comments.
